# Individual and community-level benefits of PrEP in western Kenya and South Africa: Implications for population prioritization of PrEP provision

Edinah Mudimu[1]*, Kathryn Peebles[2], Zindoga Mukandavire[3,4], Emily Nightingale[5], Monisha Sharma[2], Graham F. Medley[5], Daniel J. Klein[6], Katharine Kripke[7], Anna Bershteyn[6,8]

1 Department of Decision Sciences, University of South Africa, Pretoria, South Africa, 2 Department of Global Health, University of Washington, Seattle, WA, United States of America, 3 Centre for Data Science, Coventry University, Coventry, United Kingdom, 4 School of Computing, Electronics and Mathematics, Coventry University, Coventry, United Kingdom, 5 Department of Global Health and Development, London School of Hygiene and Tropical Medicine, London, United Kingdom, 6 Institute for Disease Modeling, Seattle, WA, United States of America, 7 Avenir Health, Washington, District of Columbia, United States of America, 8 Department of Population Health, New York University Grossman School of Medicine, NY, United States of America

☯ These authors contributed equally to this work.
* mudime@unisa.ac.za

## Abstract

### Background

Pre-exposure prophylaxis (PrEP) is highly effective in preventing HIV and has the potential to significantly impact the HIV epidemic. Given limited resources for HIV prevention, identifying PrEP provision strategies that maximize impact is critical.

### Methods

We used a stochastic individual-based network model to evaluate the direct (infections prevented among PrEP users) and indirect (infections prevented among non-PrEP users as a result of PrEP) benefits of PrEP, the person-years of PrEP required to prevent one HIV infection, and the community-level impact of providing PrEP to populations defined by gender and age in western Kenya and South Africa. We examined sensitivity of results to scale-up of antiretroviral therapy (ART) and voluntary medical male circumcision (VMMC) by comparing two scenarios: maintaining current coverage ("status quo") and rapid scale-up to meet programmatic targets ("fast-track").

### Results

The community-level impact of PrEP was greatest among women aged 15–24 due to high incidence, while PrEP use among men aged 15–24 yielded the highest proportion of indirect infections prevented in the community. These indirect infections prevented continue to increase over time (western Kenya: 0.4–5.5 (status quo); 0.4–4.9 (fast-track); South Africa:

**Data Availability Statement:** All relevant data are within the paper and its Supporting Information files.

**Funding:** Funding support was provided by the OPTIONS consortium. The OPTIONS consortium was made possible by the generous assistance from the American people through the U.S. Agency for International Development (USAID) in partnership with PEPFAR. Financial assistance was provided by USAID to Avenir Health under the terms of Cooperative Agreement No. AID-OAA-A-15-00035. The contents do not necessarily reflect the views of USAID or the United States Government. The funder provided support in the form of salaries for author [KK], but did not have any additional role in the study design, data collection and analysis, decision to publish, or preparation of the manuscript. The specific roles of these authors are articulated in the 'author contributions' section.

**Competing interests:** I have read the journal's policy and one of the authors of this manuscript has the following competing interests: KK is employed by a commercial company: Avenir Health. This does not alter our adherence to PLOS ONE policies on sharing data and materials. There are no patents, products in development or marketed products to declare.

0.5–1.8 (status quo); 0.5–3.0 (fast-track)) relative to direct infections prevented among PrEP users. The number of person-years of PrEP needed to prevent one HIV infection was lower (59 western Kenya and 69 in South Africa in the status quo scenario; 201 western Kenya and 87 in South Africa in the fast-track scenario) when PrEP was provided only to women compared with only to men over time horizons of up to 5 years, as the indirect benefits of providing PrEP to men accrue in later years.

## Conclusions

Providing PrEP to women aged 15–24 prevents the greatest number of HIV infections per person-year of PrEP, but PrEP provision for young men also provides indirect benefits to women and to the community overall. This finding supports existing policies that prioritize PrEP use for young women, while also illuminating the community-level benefits of PrEP availability for men when resources permit.

## Introduction

Sub-Saharan Africa is the region most affected by human immunodeficiency virus (HIV), accounting for more than 54% of the global population living with HIV [1]. Recent success in scaling up antiretroviral therapy (ART) coverage in the region has led to substantial declines in HIV incidence [2]. However, despite these declines, HIV incidence remains high, with approximately 1.1 million new infections in sub-Saharan Africa in 2018 [3], suggesting that additional HIV prevention methods are required to achieve epidemic control goals.

Oral pre-exposure prophylaxis (PrEP) is a promising HIV prevention method, with efficacy greater than 90% when used with high adherence [4]. The success of PrEP demonstration projects [5,6] has led to national scale-up programs in several countries in sub-Saharan Africa, including Kenya and South Africa. However, costing studies have found PrEP to be a high-cost form of HIV prevention [7,8]. Providing prevention interventions to people who are most likely to be infected increases efficiency, and young women, who experience the highest incidence, are likely to benefit the most. However, sexual networks play a key role in determining how sexually transmitted infections, such as HIV, propagate through communities [9,10]. Consequently, the degree of impact of a prevention intervention such as PrEP also depends on sexual network characteristics [11]. Providing prevention interventions to individuals with the highest network connectivity may prevent more total infections for a lower expenditure of resources [10].

We simulated PrEP provision scenarios using an individual-based network model to compare their impact, individual- and community-level benefits, and number of person-years of PrEP needed to avert one infection, using western Kenya and South Africa as case studies.

## Methods

### Model structure

We adapted an existing stochastic agent-based network model, Epidemiological MODeling software (EMOD) version 2.5, to simulate PrEP delivery for HIV prevention in western Kenya and South Africa. EMOD is an open-source model for transmission of several high-burden infectious diseases including HIV, tuberculosis, and malaria [12]. The HIV module, EMOD--HIV, is available online at https://github.com/InstituteForDiseaseModeling/EMOD alongside

a detailed model description, parameter definitions, and usage instructions are available at https://idmod.org/docs/emod/hiv/hiv-model-overview.html. Briefly, the primary mode of HIV transmission in the model is through a network of relationships categorized into three types (marital, informal, or transitory) each with different characteristic durations and age/sex patterns [13]. The networks of relationships are generated by an algorithm that tunes formation rates by age and sex for each relationship type [14,15]. Transmission risk is applied on a per-coital-act basis. Mother-to-child transmission is also included and contributes to intergenerational transmission [16].

The model was configured to represent South Africa and western Kenya using setting-specific demography (fertility by age, mortality by age and sex) and parameters with high *a priori* uncertainty (e.g., condom usage patterns, rates of concurrent sexual partnerships) were calibrated to fit age- and sex-specific HIV prevalence and incidence. We began with previously published model calibrations for western Kenya [17] and South Africa [18,19] and updated the calibrations to make use of the most recent available HIV prevalence, incidence, and intervention coverage data. The updated calibration parameters are provided in S2 Appendix S5 Table for western Kenya and S4 Table for South Africa. The final model results for prevalence of concurrent relationships are similar to published estimates [20]. We calibrated the baseline and then simulated PrEP scenarios using 250 runs of the stochastic model per scenario.

## PrEP parameterization

PrEP was implemented in EMOD-HIV as a reduction to the risk of HIV acquisition per coital act applicable to those who take up and adhere to PrEP, and time-limited to the period that these individuals are retained on PrEP [21]. In each model simulation, HIV-negative individuals in the selected population groups are eligible to enroll for PrEP. We assumed PrEP effectiveness to be 75% among adherent individuals based on estimates from oral PrEP studies [22]. PrEP continuation and re-engagement after discontinuation are not well studied. Studies available have reported high dropout rates [23,24], with many PrEP initiators discontinuing use after only 30 days [25]. In the United States, PrEP continuation has been reported to last an average of 14.5 months, with women and young adults (18 to 24 years of age) having the shortest continuation of approximately 6.9 and 8.6 months, respectively [24]. Based on these studies, we modelled duration of PrEP use with a Gaussian distribution with a mean of 180 days and a standard deviation of 30 days. We used an optimistic mean duration of 180 days on PrEP, assuming that future innovations to overcome barriers to PrEP uptake and continuation in implementation programs in Africa will favour longer durations of PrEP use [26] than currently observed [27]. Following PrEP discontinuation, we assumed individuals would not re-initiate PrEP for a minimum of six months. We assumed that PrEP in our model became available in 2016 for both western Kenya and South Africa, with 45% PrEP coverage among HIV-negative individuals in the corresponding age/sex category throughout the simulation period. This high coverage of PrEP provided more stable numbers of individuals on PrEP in our stochastic model, allowing us to evaluate community-level benefits of PrEP.

## Model scenarios

We evaluated strategies providing PrEP to different populations defined by age (15–24 and 15–49) and sex (PrEP provided either to women or to men, but never to both). Each PrEP strategy was evaluated in the context of either a pessimistic (status quo) or optimistic (fast-track) baseline of ART and male circumcision scale-up (Table 1). The status quo baseline assumes continuation of current rates of uptake of HIV services (other than PrEP), while the fast-track baseline assumes achievement of programmatic targets. Specifically, in the fast-track

**Table 1. Status quo and Fast track scenarios and PrEP uptake assumptions.**

| Scenarios | ART | | Circumcision | |
|---|---|---|---|---|
| | **Kenya** | **South Africa** | **Kenya** | **South Africa** |
| Status quo | 2016 ART uptake[a] | 2015 ART uptake[b] | | 2010–2017 coverage |
| Fast-track | 95-95-95 targets[c] | 95-95-95 targets[c] | 95% coverage | 2018: 650,000[d] |
| | | | | 2019: 600,000 |
| | | | | 2020: 550,000 |
| | | | | 2021: 500,000 |
| | | | | 2022–2036: 100,000 |

| PrEP uptake assumptions | | |
|---|---|---|
| **Parameter** | **Value** | **Reference** |
| Efficacy | 0.75 | Baeten JM et al. 2012 [23] |
| Coverage | 0.45 | Assumption |
| Duration on PrEP | Gaussian(180,30) days | Assumption |
| Re-engagement | 6 months | Assumption |

[a] Kenya 2016 ART uptake guidelines [31].

[b] South Africa 2015 ART uptake guidelines [32].

[c] United Nations AIDS. Fast-Track: ending the AIDS epidemic by 2030 [28].

[d] South Africa 2017–2021 National Strategic Plan targets [29].

baseline, ART coverage is assumed to reach UNAIDS 95-95-95 targets for 2030 [28]. Circumcision is assumed to reach South Africa 2017–2021 National Strategic Plan targets [29] and 100,000 cicumcisions yearly from 2022–2036 (to maintain 80% coverage [30]) and in western Kenya, to achieve 95% coverage among men ages 15–24 years by 2020.

For each PrEP provision strategy, we evaluated the number of HIV infections averted directly and indirectly by PrEP over a 20-year time horizon. To disentangle the direct (infections prevented among PrEP users) and indirect (infections prevented among non-PrEP users as a result of PrEP) effects of PrEP, we calculated the ratio of indirect-to-direct infections averted, which provides an estimate of the number of indirect infections averted for every one direct infection averted. We calculated this quantity by first randomizing simulated persons to receive either placebo or active PrEP. The direct protective effect was then calculated as the difference between new infections among placebo recipients and active PrEP recipients. The indirect effect was calculated as the difference in total new infections between the counterfactual simulation (no PrEP in the community) and intervention scenarios, less infections averted by direct PrEP effects. We also calculated the number of person-years of PrEP required to avert one infection, regardless of whether the mechanism of protection is direct or indirect.

## Results

### Averted HIV infections

In the status quo scenario in which ART and male circumcision coverage was maintained at present-day levels, PrEP use among women yielded the greatest number of total averted infections (Fig 1) in the initial years of the PrEP program. However, PrEP use among men averted more infections than PrEP use among women after eight years (in scenarios where PrEP provision is directed to those aged 15–24) or four years (PrEP to those aged 15–49). In fast-track scenarios, on the other hand, the greatest number of infections averted is highest when offered to women over all time horizons, with the exception of the broad age target of individuals aged 14–49 years in South Africa, where there was no difference by gender. In the status quo

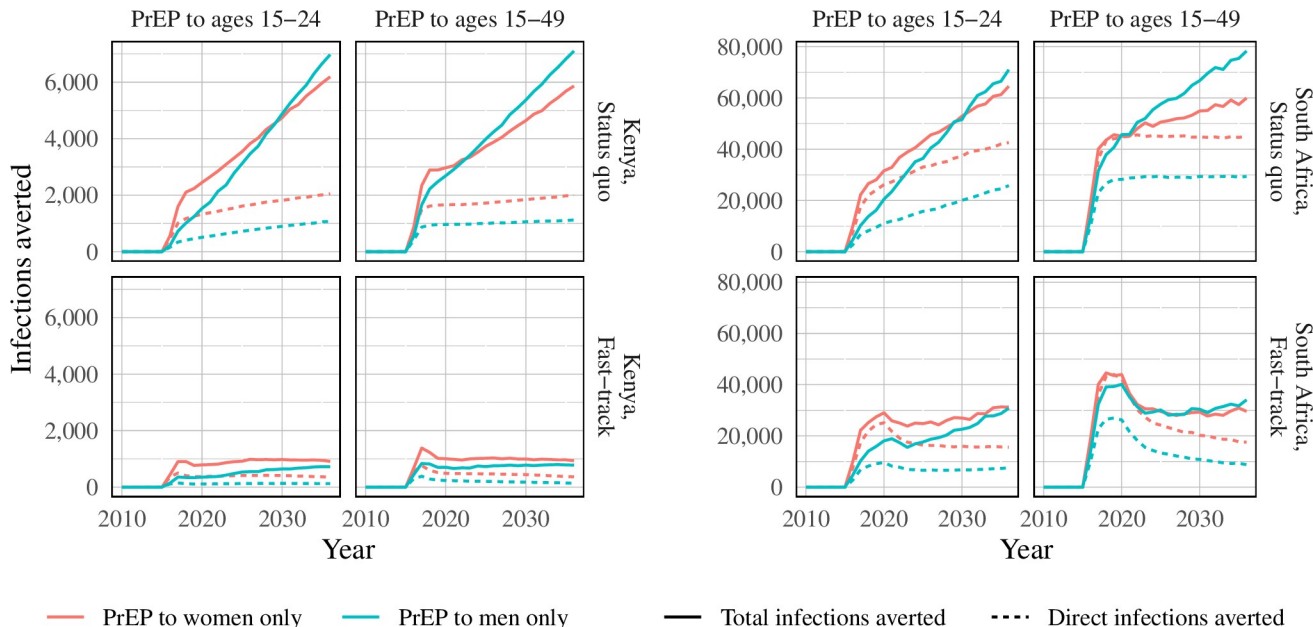

**Fig 1. Direct prevention and total averted infections compared to status quo and fast-track baseline scenarios in South Africa and a status quo baseline scenario in Kenya.**

scenarios, infections averted by PrEP continue to increase over time, whereas with concurrent scale-up of ART and circumcision, the epidemic impact of PrEP is limited to the five-year period following rollout due to rapid declines in incidence driven by achievement of programmatic targets in ART and male circumcision coverage.

## Indirect-to-direct prevention ratio

The indirect-to-direct prevention ratio was highest when PrEP provision was directed to individuals aged 15–24 years of age in both western Kenya and South Africa over all time horizons (Fig 2). Indirect-to-direct ratios increased over time. Providing PrEP to men yielded the highest indirect-to-direct ratio (western Kenya: 0.4–5.5 (status quo); 0.4–4.9 (fast-track); range in South Africa: 0.5–1.8 (status quo); 0.5–3.0 (fast-track)); the first number in the given indirect-to-direct ratio intervals represent the beginning of the time horizon (2016) and the second number represent the end of the time horizon (2036). A similar pattern was observed in scenarios where PrEP was delivered concurrently with scale-up of ART and circumcision to meet fast-track targets, though the ratios were lower in the fast-track scenarios.

## Number of person-years of PrEP needed to prevent one HIV infection

In status quo scenarios, 59 person-years of PrEP for women aged 15–24 in western Kenya and 69 person-years of PrEP for women aged 15–24 in South Africa were required to avert one HIV infection over a 5-year timeframe (Fig 3). Over the same time horizon, approximately 94 person-years of PrEP for men in western Kenya and 115 person-years of PrEP for men in South Africa are required to avert one infection. In western Kenya, the number of person-years of PrEP needed to avert one infection quadrupled in the fast-track scenario (from 59 to 209 for women and from 94 to 490 for men). In contrast, there was a smaller increase in the number of person-years of PrEP required to avert one infection in fast-track scenarios in South Africa (from 69 to 90 for women and from 115 to 120 for men). In both countries, over

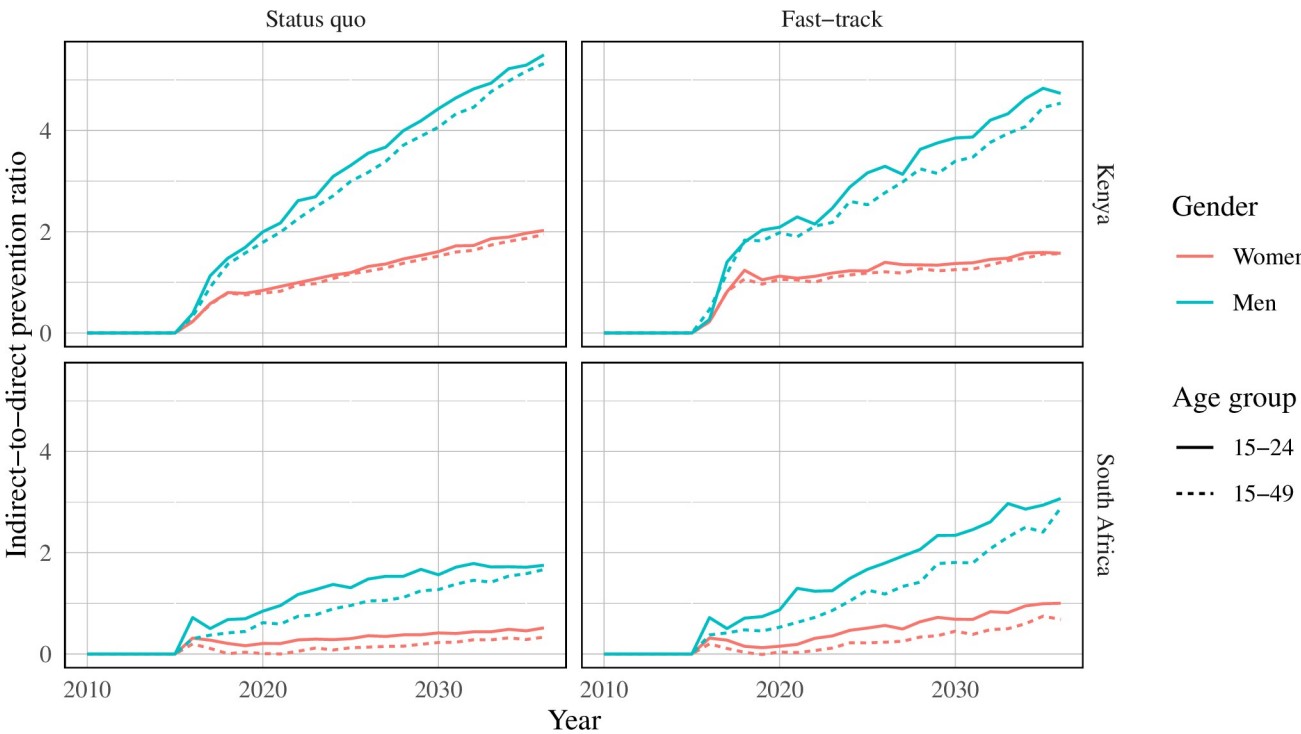

**Fig 2. Indirect-to-direct ratio of infections averted in South Africa and Kenya.** Scenarios show results from PrEP provision by age and gender.

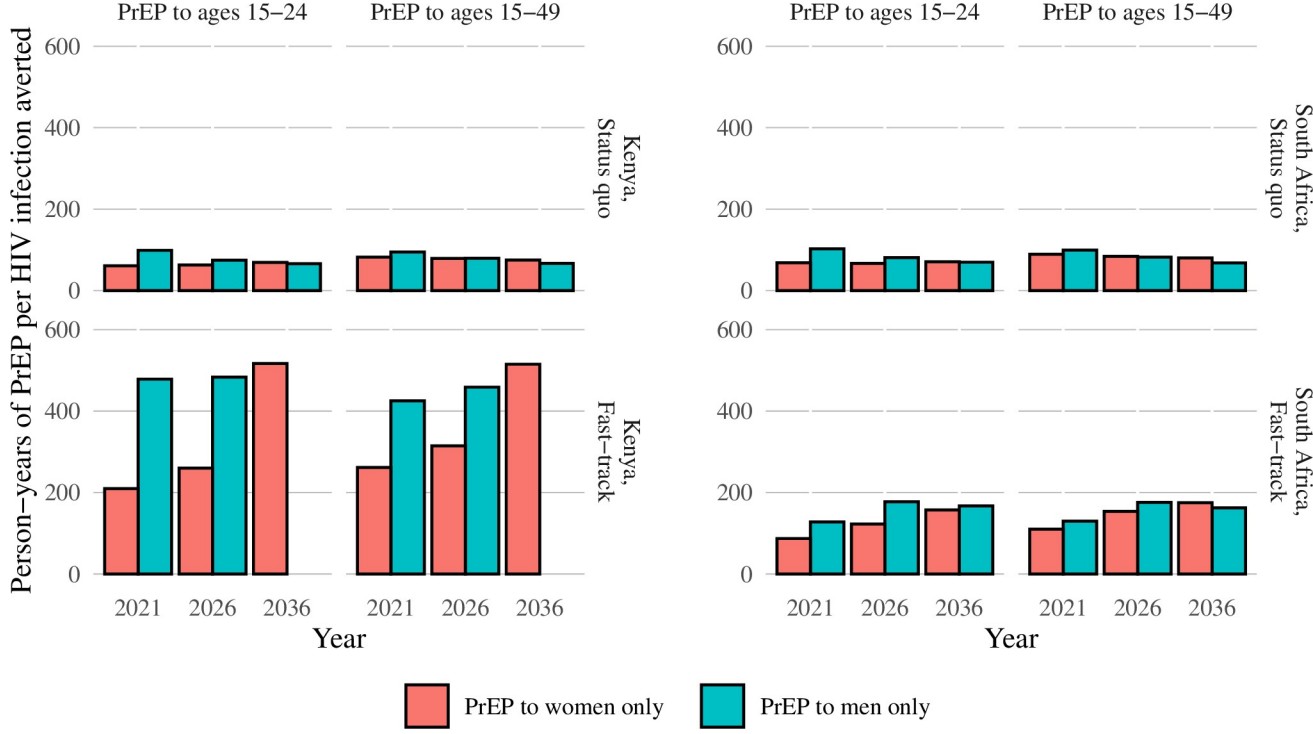

**Fig 3. Number of person-years of PrEP needed to prevent one HIV infection by country, baseline scenario, and PrEP provision strategy.**

5- and 10-year time horizons, giving PrEP to women aged 15–24 required the least resources to avert one infection. Over a 20-year time horizon in status quo scenarios, the secondary benefits of offering PrEP to men accrue sufficiently that a smaller number of person-years of PrEP are needed to avert one infection. In fast-track scenarios, offering PrEP to women required fewer person-years of PrEP to avert one infection across all time horizons in Kenya, and a similar number of person-years of PrEP given to men or women were required in South Africa to avert one infection (Fig 3).

## Discussion

We modeled the potential impact of PrEP provision strategies in combination with both status quo and fast-track scale-up of ART and male circumcision in two countries in sub-Saharan Africa with varying HIV epidemics. In status quo scenarios in both countries, providing PrEP to women resulted in the greatest number of HIV infections averted within a 4- to 8-year time horizon. Long-term impacts (after 8 years from the beginning of PrEP provision) are greater when providing PrEP to men. However, as countries scale up ART and male circumcision to meet UNAIDS 95-95-95 targets, our analyses suggest that offering PrEP to women will result in both the greatest number of infections averted and the lowest person-years of PrEP use per infection averted. In both status quo and fast-track scenarios, we observed that providing PrEP to men resulted in the highest indirect-to-direct prevention ratio. This is likely because, in our model, men have more sexual partners and greater partner concurrency resulting in higher network connectivity. However, the benefits of providing PrEP to men tended to accrue in later years, and preventing direct infections in women offered the greatest impact in the first five years in both status quo and fast-track scenarios. These results were similar in the distinct epidemiological settings of both Kenya and South Africa, supporting global recommendations that prioritize PrEP for young women.

We utilized a well-established network model and presented average model results across a set of 250 simulations per scenario. As the impact of PrEP on the HIV epidemic depends on the network characteristics of not just the PrEP recipients, but also their partners, and their partners' partners, the use of a network-based model allows for realistic estimation of community-level health benefits. Previous mathematical modelling [33] demonstrated that earmarking prevention interventions to key population groups like female sex workers can significantly reduce HIV incidence, but large-scale impact in mature epidemics will require interventions for both key and non-key populations [34]. Indeed, our analyses showed the impact for a given set of resources may still be high when directed to a less connected, but more at-risk, population group, highlighting the importance of providing PrEP to general population groups at elevated risk.

Our study has several limitations. Firstly, PrEP delivery at scale is a relatively recent addition to the HIV prevention toolkit, and data from real-world settings on key characteristics of PrEP use are limited. Consequently, we made several simplifying assumptions with respect to PrEP continuation, re-engagement in care (cycling on and off PrEP), and efficacy. PrEP efficacy was assumed to be equal between men and women and across age groups. This assumption has a direct effect on the community-level impact of allocating PrEP to different groups. For example, allocating PrEP to a group with low adherence would be less effective at the community-level, including indirect effects in later years. However, although the absolute health benefits of different PrEP strategies would change under different assumptions for PrEP efficacy, uptake and retention, the relative benefits of the strategies should remain the same, with PrEP for women yielding the greatest number of HIV infections averted. For this reason, we compared extreme scenarios (provision of PrEP to individuals of just a single age/sex group) to enable comparison

of the impact of PrEP under each strategy, with the expectation that programs will provide PrEP to a mix of population groups informed by the impact in each group as well as the accessibility, adherence, and demand for PrEP in each group. Secondly, PrEP enrollment in the model does not depend on the risk profile of the individuals. In real-world settings, PrEP is recommended to those at substantial risk of acquiring HIV, which includes characteristics such as having a recent sexually transmitted infection; beliefs about a partner's HIV status or viral suppression status; residing in a transmission hotspot; or other behavioral, biological, or demographic indicators of HIV risk. Future work in this area would benefit from detailed modeling of individual risk profiles to inform PrEP provision strategies at a micro-level. Identifying PrEP provisions strategies that maximise impact would improve the value proposition of PrEP.

Our analyses evaluated the number of person-years of PrEP required to prevent one HIV infection as a proxy for the efficiency of providing PrEP to different populations based on sex and age. Future analyses may extend this work with evaluations of offering PrEP to persons at substantial risk of HIV acquisiation, regardless of gender or age and evaluations of the economic impact and cost-effectiveness of distributing PrEP Primary cost data for exact costs of PrEP delivery, which may vary based on drug costs and delivery models, is limited [35], with even less data available on costs of PrEP ancillary services. However, our analysis provides insight into resource allocation by estimating the community-level impact of different PrEP provision strategies to identify the strategy with the highest health benefits.

## Conclusions

Providing PrEP to women aged 15–24 prevents the greatest number of HIV infections for a given amount of resources over most time horizons due to women's higher HIV risk, though the indirect benefits of providing PrEP to only men were highest across all time points due to their higher rate of transmission. This finding supports existing policies that prioritize PrEP provision for young women.

## Supporting information

**S1 Appendix. Model-fitting parameters western Kenya.**
(XLSX)

**S2 Appendix. Model-fitting parameters South Africa.**
(XLSX)

## Acknowledgments

We gratefully acknowledge the developers of the EMOD disease modeling framework and its associated scripts and calibration capabilities, especially Daniel Bridenbecker and Clark Kirkman IV.

## Author Contributions

**Conceptualization:** Edinah Mudimu, Kathryn Peebles, Daniel J. Klein, Katharine Kripke, Anna Bershteyn.

**Formal analysis:** Edinah Mudimu, Kathryn Peebles, Zindoga Mukandavire, Emily Nightingale, Monisha Sharma, Graham F. Medley, Daniel J. Klein, Katharine Kripke, Anna Bershteyn.

**Software:** Edinah Mudimu, Kathryn Peebles, Daniel J. Klein, Anna Bershteyn.

**Writing – original draft:** Edinah Mudimu, Kathryn Peebles.

**Writing – review & editing:** Edinah Mudimu, Kathryn Peebles, Zindoga Mukandavire, Emily Nightingale, Monisha Sharma, Graham F. Medley, Daniel J. Klein, Katharine Kripke, Anna Bershteyn.

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
