## [Decision Letter · Decision Letter 0]

23 Sep 2020

PONE-D-20-24256

Individual and community-level benefits of PrEP in western Kenya and South Africa: implications for population prioritization of PrEP Provision

PLOS ONE

Dear Dr. Mudimu,

Thank you for submitting your manuscript to PLOS ONE. After careful consideration, we feel that it has merit but does not fully meet PLOS ONE’s publication criteria as it currently stands. Therefore, we invite you to submit a revised version of the manuscript that addresses the points raised during the review process.

Please address all of the reviewers' comments in your revision. It will be particularly important to address the comments about describing your methods in greater detail and about reporting more granular results (e.g., in tables).

We look forward to receiving your revised manuscript.

Kind regards,

Douglas S. Krakower, MD

Academic Editor

PLOS ONE

Journal Requirements:

We note that one or more of the authors are employed by a commercial company: Avenir Health.

2.1. Please provide an amended Funding Statement declaring this commercial affiliation, as well as a statement regarding the Role of Funders in your study. If the funding organization did not play a role in the study design, data collection and analysis, decision to publish, or preparation of the manuscript and only provided financial support in the form of authors' salaries and/or research materials, please review your statements relating to the author contributions, and ensure you have specifically and accurately indicated the role(s) that these authors had in your study. You can update author roles in the Author Contributions section of the online submission form.

2.2. Please also provide an updated Competing Interests Statement declaring this commercial affiliation along with any other relevant declarations relating to employment, consultancy, patents, products in development, or marketed products, etc.  

Reviewers' comments:

Reviewer's Responses to Questions

**Comments to the Author**

1. Is the manuscript technically sound, and do the data support the conclusions?

Reviewer #1: Partly

Reviewer #2: Yes

2. Has the statistical analysis been performed appropriately and rigorously? 

Reviewer #1: Yes

Reviewer #2: Yes

3. Have the authors made all data underlying the findings in their manuscript fully available?

Reviewer #1: No

Reviewer #2: Yes

4. Is the manuscript presented in an intelligible fashion and written in standard English?

Reviewer #1: Yes

Reviewer #2: Yes

5. Review Comments to the Author

Reviewer #1: This paper presents an agent-based network model of HIV transmission among heterosexuals in Western Kenya and South Africa. The study compares the impact of introducing PrEP to either men or women, with variable scenarios based on age group provision and exogenous processes (i.e., increased ART and male circumcision) across varying time periods. The authors found that PrEP provision to women had the greatest immediate impact and was also the most efficient allocation of resources.

Overall, the study is well-designed and relies on a network model which has been used in other studies. The model complexity is appropriate for this type of study. I appreciate that the authors present a number of counterfactual scenarios which could be helpful for policy makers. The manuscript is timely, as resources for public health may be limited (i.e., due to coronavirus) so policy makes may need to be strategic about allocating resources for other priorities. However, the overall presentation of the methods and results should be improved before the manuscript is ready for publication.

Major points:

1. The methods section does not provide sufficient information for a model of this complexity. The authors point to some references, but it was not immediately obvious which references were most relevant. I suggest the authors rework the methods section to provide greater detail about the overall structure of the model and some key parameters which are relevant to the research question. If word count is a problem, it is common for models of this complexity to have a technical appendix. One of the references (13) was helpful to answer some of my questions, but this was published in 2012 so it is not clear what, if anything, has changed about the model and code since then. It may also be helpful to remove the references to your group’s other studies (refs 11 and 12) using this model and just point the readers to the reference that is clearly describing the model design. If the other references are relevant to piece together changes to the model since 2012, then I think it would be most appropriate to compile a single document (i.e., technical appendix) which clearly describes the current model. Reference 10 is too general to provide much relevant information for the present study.

2. The sexual network characteristics seems to be the most important information that was left out, since this is the underlying mechanism driving the model results. Summary information about concurrency and assortativity should be provided so that readers can better understand differences between men and women (i.e., degree, concurrency and anything else that is relevant) and how this is generating the model results. I don’t think it is necessary to have all parameters summarized in the main text of the manuscript, as this would obviously be too much information. I know that some of this information is listed in the excel files provided, which I appreciate, but these are summarizing the calibration so it is unclear to me what the final numbers were and whether there is a single calibrated parameter for each that is consistent across counterfactual scenarios. It would be really helpful to summarize specific numbers as well as words in the manuscript.

3. The results section was challenging to follow. There are many counterfactual scenarios presented and it was difficult to keep track of each. Additionally, the authors present the results qualitatively, rather than quantitatively, for some sections. I suggest the authors add outcome measures throughout; ideally, some kind of a point estimate, such as a median, as well as an interval, such as a 95% simulation interval.

4. There are no tables presenting results of any kind, which I was surprised to see. This would be helpful to summarize the results, but also could serve as a way to guide readers through the various counterfactual comparisons that are presented and how the outcomes increase/decrease across varying conditions.

Minor issues:

Abstract:

1. The Results section presents no quantitative results of any kind, so there is little difference in what is written in the Results section vs. Conclusions. Also consider re-writing this section so that it is clearer what the counterfactual contrasts are which are producing the results. (I know abstracts are short and appreciate that this is challenging).

2. Line 40: I’m a little confused by the last sentence in this section, as my interpretation of the results is that total infections averted varied based on time (prioritizing immediate results) and assumptions about fast-track vs status quo.

3. The Conclusions (e.g., first sentence, line 43) are clearer than what is written in the Results section, although both are saying essentially the same thing.

Introduction:

4. Line 50: The 54% number is ambiguous as written. Does this refer to 54% of new infections? Overall prevalence?

5. Line 52: I certainly don't disagree that incidence is unacceptably high, however this implies that there is some threshold for which ongoing transmission would be acceptable. Perhaps remove the word “unacceptably”?

6. Lines 63-64: Some references for the sentence “However, sexual networks play…” would be helpful since not all readers will be familiar with network science and models. Two suggestions (though there are others that would be appropriate): Rothenberg "How a net works", STD, 2001; and Liljeros "Sexual networks: implications for the transmission of sexually transmitted infections" Microbes and Infection, 2003.

7. Line 66-67: Again, a reference here would be helpful. The two above may be relevant. Also, Morris has some relevant references (e.g., "Concurrent partnerships and the spread of HIV," AIDS, 1997).

Methods:

8. Line 76: I’m glad to see a link to the github repository. However, is there a specific repository for the code used for the present study?

9. Line 77: The idmod.org link is broken. I found this page https://idmod.org/docs/emod/hiv/hiv-model-overview.html which I think is relevant and helpful, but doesn’t provide sufficient information for how this specific study was implemented.

10. Line 78: What is the age structure of the model? When do individuals enter the model? Do they leave at age 49? Or is that just the cutoff for PrEP eligibility?

11. Line 79: How is transmission operationalized? Condoms are not mentioned in the manuscript at all. How does ART impact transmission probabilities?

12. PrEP Parameterization: Were any other aspects of the PrEP clinical protocol modeled? What was the frequency of HIV screening? Was this the same as individuals not using PrEP? Does knowledge of HIV infection (via screening and diagnosis) change behavior (e.g., condom use) in the model? What about initiation of PrEP or ART?

13. Line 91: It seems PrEP effectiveness was the same for all individuals using PrEP. Was there any heterogeneity in how this was parameterized (e.g., operationalized via varying adherence)? Is adherence known to vary between men and women (or by age?) in this population? This seems like an important limitation that should be either addressed in the model or discussed.

14. Line 92: Why is long-acting PrEP mentioned? Is this just to say that the model was limited to oral PrEP?

15. Line 106: Does 45% coverage refer to 45% of all women in scenarios targeting PrEP to women? Or is this 45% among the women of a specific age group? (Same questions for men).

16. Line 130: The time ranges are listed as 10 or 20 year horizons, but the results refer to 4 year, 8 year (possibly other) timelines. My suspicion is that the authors ran models for 10 or 20 years, but looked at results across other ranges, which is fine technically, but is confusing for how the results are presented. It would be helpful to provide clarity.

17. Line 136: The description for how indirect effects were calculated is a little confusing as written. Does infections averted by direct protection refer to the quantity calculated for direct effects?

18. Line 137: I believe this is the first time that the reference scenario (no PrEP) has been mentioned. It would be helpful to mention this earlier and describe how the reference scenario is used.

19. Line 139: For person-years required to prevent one infection, is this total effects? Only direct effects?

Results:

20. Figures: Are there figure legends? I also noticed in Figure 3 that the scale on the y-axis is not the same across all plots, so the figure is misleading if the reader doesn’t look closely. The image quality is poor, so it was difficult to view the figures; this may be on the way the PDF is generated by Plos One, but make sure the image quality is sufficient.

21. Line 145: By changing the age range that is targeted, does this imply more individuals are using PrEP? I.e., if it was 45% in men 15-24 and then 45% in men 15-49 (and similarly for women). Or is it the same number using PrEP but spread across a wider age range?

22. Line 159: Are these 95% simulation intervals? It would be helpful to have a median value here since the interval seems wide and ratios less than 1 have a different interpretation than ratios above 1.

Discussion

23. Line 206: “Good-fitting parameter sets” is jargon that isn’t previously defined. If this is relevant for the Discussion, I suggest using plain language so readers without modeling experience can understand what this means.

24. Line 212: This conclusion seems misleading, since PrEP was not targeted based on degree or concurrency specifically, but allocated based on sex and age. These variables may correlate with degree and concurrency, but within groups PrEP was allocated equally.

Reviewer #2: This interesting manuscript addresses an important question – prioritization of offer of PrEP to different populations in western Kenya and South Africa in order to maximize impact in reducing new HIV infections. Overall, the manuscript is well written and provides important data to guide policy. I have several questions and suggestions for the authors to consider to strengthen the manuscript.

Major comments:

1) Because some national guidelines (e.g. in Kenya) now allow for offer of PrEP to persons at substantial risk of HIV acquisition, regardless of gender, it would be helpful to include an analysis in which PrEP is offered to both men and women at substantial risk. The current analyses in which PrEP is offered only to women or only to men are interesting exercises but are limited in terms of applicability to current programs. Although young women should understandably be prioritized for PrEP based on high incidence, men should also be prioritized for PrEP (as the authors note) to reduce incidence among women/their communities and, as many would argue, for their own health. Thus, the manuscript would be greatly strengthened by including an analysis in which PrEP is offered to both men and women at substantial risk.

2) Could the authors please provide additional information on expected ART and VMMC coverage within the manuscript text and in Table 1 under different scenarios? E.g. when assuming status quo, do the guidelines referenced recommend ART treatment for all or based on CD4 thresholds, and what levels of ART coverage could that result in? How is ART coverage expected to differ by sex? This would help to contextualize the results.

3) In the methods would be helpful to define “PrEP coverage” – is the denominator persons at substantial risk? All HIV-noninfected persons? If the latter, would be helpful to clarify this before Discussion.

4) Community-level and population-level impact seem to be used at different times – could the authors kindly define these terms and if they are intended to be the same or different?

5) Given that the sexual network parameters impact the results significantly, it would be helpful to provide more information on network assumptions in the methods section

Minor comments:

1) Could the authors also please provide more background on HIV prevalence and incidence in western Kenya and South Africa for readers with less knowledge of these regions?

2) Introduction notes that PrEP is a relatively high-cost intervention but discussion notes that costing data are limited – could the authors clarify what data are cited to indicate high cost of PrEP (noting that this may vary based on drug costs, personnel, etc)?

3) Methods – based on recent data (e.g. from AIDS2020) many individuals cycle on and off PrEP frequently. Could the authors comment on the impact of cycling back on PrEP in a shorter time than 6 months?

6. PLOS authors have the option to publish the peer review history of their article (what does this mean?). If published, this will include your full peer review and any attached files.

Reviewer #1: **Yes: **Kevin Maloney

Reviewer #2: No

---

## [Author Response · Author response to Decision Letter 0]

13 Oct 2020

We have substantially revised the methods and provide additional clarity on what EMOD is, the transmission modalities in the EMOD-HIV model and the procedures and parameters used to calibrate the model to the two settings in our analysis. A reference that describes the technical implementation of the sexual network in the EMOD-HIV model has been added. Detailed feedback is included in the Reponses to reviewers comments letter.

---

## [Decision Letter · Decision Letter 1]

17 Nov 2020

PONE-D-20-24256R1

Individual and community-level benefits of PrEP in western Kenya and South Africa: implications for population prioritization of PrEP Provision

PLOS ONE

Dear Dr. Mudimu,

Thank you for submitting your manuscript to PLOS ONE. After careful consideration, we feel that it has merit but does not fully meet PLOS ONE’s publication criteria as it currently stands. Therefore, we invite you to submit a revised version of the manuscript that addresses the points raised during the review process.

Please attend the these minor revisions, which will be required and sufficient for publication in PLOS One. Thank you for your careful attention to these details.

We look forward to receiving your revised manuscript.

Kind regards,

Douglas S. Krakower, MD

Academic Editor

PLOS ONE

Reviewers' comments:

Reviewer's Responses to Questions

**Comments to the Author**

1. If the authors have adequately addressed your comments raised in a previous round of review and you feel that this manuscript is now acceptable for publication, you may indicate that here to bypass the “Comments to the Author” section, enter your conflict of interest statement in the “Confidential to Editor” section, and submit your "Accept" recommendation.

Reviewer #1: (No Response)

Reviewer #2: All comments have been addressed

2. Is the manuscript technically sound, and do the data support the conclusions?

Reviewer #1: Yes

Reviewer #2: Yes

3. Has the statistical analysis been performed appropriately and rigorously? 

Reviewer #1: Yes

Reviewer #2: Yes

4. Have the authors made all data underlying the findings in their manuscript fully available?

Reviewer #1: Yes

Reviewer #2: Yes

5. Is the manuscript presented in an intelligible fashion and written in standard English?

Reviewer #1: Yes

Reviewer #2: Yes

6. Review Comments to the Author

Reviewer #1: The authors have greatly improved the overall readability of the manuscript and clarity of the methods. My major concerns have been addressed and I only have a few minor comments at this stage. In addition, the manuscript would benefit from a careful edit for spelling and typos, as there are some minor mistakes throughout and I don’t think Plos One provides copy editing.

1. The authors provided additional clarity about how PrEP effectiveness is operationalized in their model (i.e., 75% on average), which I appreciate. However, it remains unresolved that effectiveness is assumed to be equal between men and women, and between age groups. This is an important limitation, given that the primary study question is evaluating the population impact of allocating PrEP to different groups. For example, if adherence is lower among men (and thus the average effectiveness of PrEP is lower) then allocating PrEP to men may be far less effective on a population level, including indirect effects in later years, than projected. The authors need to spend more time discussing this limitation.

2. Minor comment: Line 175 – I may have missed it, but it is still unclear what this interval represents. I appreciate the explanation in the response to my original question, but that’s not how I read it in the manuscript. On line 174, it says the ratios increase overtime, but it’s not clearly explained that the first number in the interval represents the beginning of the time horizon and the second number represents the end of the time horizon.

3. Minor comment: Line 223 – this says 250 simulations, but line 100 in the methods indicates 300 simulations.

Reviewer #2: (No Response)

7. PLOS authors have the option to publish the peer review history of their article (what does this mean?). If published, this will include your full peer review and any attached files.

Reviewer #1: **Yes: **Kevin Maloney

Reviewer #2: No

---

## [Author Response · Author response to Decision Letter 1]

8 Dec 2020

We have addressed the minor revisions from our reviewers. See attached responses to reviewers document.

---

## [Editor Report · Decision Letter 2]

16 Dec 2020

Individual and community-level benefits of PrEP in western Kenya and South Africa: implications for population prioritization of PrEP Provision

PONE-D-20-24256R2

Dear Dr. Mudimu,

We’re pleased to inform you that your manuscript has been judged scientifically suitable for publication and will be formally accepted for publication once it meets all outstanding technical requirements.

Kind regards,

Douglas S. Krakower, MD

Academic Editor

PLOS ONE
---

## [Editor Report · Acceptance letter]

22 Dec 2020

PONE-D-20-24256R2 

Individual and community-level benefits of PrEP in western Kenya and South Africa: implications for population prioritization of PrEP provision 

Dear Dr. Mudimu:

I'm pleased to inform you that your manuscript has been deemed suitable for publication in PLOS ONE. Congratulations! Your manuscript is now with our production department. 

Kind regards, 

on behalf of

Dr. Douglas S. Krakower 

Academic Editor

PLOS ONE